# Bulk Polymerization of Thermoplastic Shape Memory Epoxy Polymer for Recycling Applications

**DOI:** 10.3390/polym15040809

**Published:** 2023-02-06

**Authors:** Haitao Zhuo, Zhen He, Jun Liu, Guocheng Ma, Zhenghe Ren, Youhan Zeng, Shaojun Chen

**Affiliations:** 1College of Chemistry and Environmental Engineering, Shenzhen University, Shenzhen 518060, China; 2College of Materials Science and Engineering, Shenzhen University, Shenzhen 518053, China

**Keywords:** shape memory, epoxy resin, bulk polymerization, recycling, thermal-plastic

## Abstract

Conventional epoxy polymers are thermo-set and difficult to recycle and reuse. In this study, a series of linear thermoplastic epoxy polymers (EPx) with shape memory properties were prepared by using a bifunctional monoamine diglycolamine (DGA) as a curing agent and an equivalent amount of bifunctional rigid epoxy resin (E-51) and bifunctional flexible epoxy resin (polypropylenglycol diglycidyl ether, PPGDGE) in a bulk polymerization reaction. The results showed that these samples can fully react under the curing process of, 60 °C/2 h, followed by 80 °C/2 h, followed by 120 °C/2 h. The introduction of different contents of PPGDGE can adjust the T_g_ of the material to adapt to different environmental requirements, and can significantly increase the fracture strain of the material and improve its micro-phase separation structure. Thus, R_f_ of the material is close to 100%, and R_r_ is increased from 87.98% to 97.76%. Importantly, this linear chain structure allows the material to be easily recycled and reprocessed by dissolving or melting, and also means the material shows potential for 3D printing or other thermoplastic remolding.

## 1. Introduction

Since Charles’s accidental discovery of the shape memory properties of chemically cross-linked polyethylene in 1960, shape memory polymer (SMP) types have been developed in polyolefin, norbornene, polyurethane, polylactic acid, polyacrylate, epoxy, and other systems that are used in cable protection, biomedical, aerospace, mechanical, and other applications [1,2]. SMPs are a class of smart materials that have a permanent shape and can be automatically reverted to this original permanent shape by shape editing under the action of specific stimuli [3] such as heat [4,5], a magnetic field [6], electricity [7], light [8,9,10], pH [11], solution [12], solvent [13], etc. SMPs have the advantages of adjustable transition temperatures, controllable properties, being light weight, and active deformation, which have gained them a lot of attention. Due to the unique properties of SMPs that enable their use in 3D printed structures, they have shown important application potential and practical value in the fields of smart devices [14], soft robots [15], hinges [16], grippers [17], origami [18], bone scaffolds [19], and stents [20].

While SMPs have been developed in a number of functionalized systems, and are widely used in many fields, the range itself is quite limited, focusing on thermoplastics such as thermoplastic polyurethane [21,22], PETG [23], polylactic acid, or polyethylene and light-curing acrylates. They have the advantages of simple preparation methods, wide application areas, and being recyclable, but more types of these materials still need to be developed to meet the needs of different applications. A variety of these materials have certain disadvantages such as large volume shrinkage after curing, poor thermal stabilities, and strong irritating odors, which severely limit the further development of 3D printed SMPs. Compared to other shape memory materials, shape memory epoxy polymer (SMEP) has many excellent physicochemical properties, such as a good thermal stability, high mechanical strength, good environmental corrosion resistance, low volume shrinkage after curing, and excellent processing properties, and can be intelligently converted to permanent and temporary shapes by external stimuli. It has a wide range of research and applications in structural parts, adhesives, and aerospace [24]. However, the majority of SMEPs in the current study are cross-linked in a three-dimensional network structure, leading to a general problem of low fracture strain in their mechanical properties. They have been limited to the traditional casting and forming process in terms of manufacturing and processing, which limits the further development and application of the material. What’s more, the permanent cross-linked network structure of traditional epoxy resins after curing makes them impossible to reprocess, repair, and dissolve, making the recycling of epoxy resin products a huge challenge.

Unlike conventional thermoset SMEPs, thermoplastic SMEPs have good recyclability, repairing, and shape memory properties [25], and can be used in a variety of applications, such as 4D printing [21,22,23], smart coatings [26], and smart textiles [27]. Thus, thermoplastic SMEPs have become attractive to researchers in recent years. Niet al. prepared thermoplastic epoxy polymers (EP-TP) by a polymerization reaction using epoxy resin blends consisting of epoxy resin and phenol monomers; the first shape memory thermoplastic epoxy filament (SMEF-TP) was successfully developed by a melt drawing process, and has excellent shape memory properties, with a shape fixation rate of 97%, and a shape recovery rate of over 97%. Lu et al. [28] replaced conventional epoxy resin curing agents with 4-aminophenyl disulfide, containing disulfide bonds, and compounded aqueous epoxy resin (sWEP) with polydopamine (PDA) nanoparticles, with the photothermal conversion effect by melt mixing and hot pressing, the PDA/sWEP composite exhibited a shape memory fixation rate of approximately 97.4% and a shape memory recovery rate of approximately 67.8%. Chen et al. [29] adjusted the properties and functions of a pristine TPU that contained disulfide bonds in the main chains and intramolecular hydrogen bonding, by blending reversible epoxy domains, as well as identifying that the thermal plasticity for pristine TPU and its blends is due to the exchanging bonds [30].

In this study, a novel thermoplastic shape memory epoxy resin with high thermal stability and low shrinkage was prepared with epoxy resin, polypropylene glycol diglycidyl ether (PPGDGE), and diethylene glycolamine (DGA). The relationship between the EPx structure and performance is systematically analyzed and the potential of the material for use in 3D printing is initially explored.

## 2. Materials and Methods

### 2.1. Materials

The materials were three commercially available polymers. Epoxide resin (E-51) was purchased from MACKLIN Biochemical Technology Co., Ltd. (Shanghai, China); polypropylene glycol diglycidyl ether (PPGDGE) was obtained from Aladdin Biochemical Technology Co., Ltd. (Shanghai, China); diethylene glycolamine (DGA) was purchased from Aladdin Biochemical Technology Co., Ltd. (Shanghai, China).

### 2.2. Preparation of Thermoplastic SMEPs

In this study, thermoplastic SMEPs were prepared from rigid structured bisphenol A epoxy resin E-51 (relative molecular mass of about 392 g/mol), flexible structured PPGDGE (relative molecular mass of about 625 g/mol), and monoamine curing agent DGA. The synthetic route is shown in Figure 1, and the detailed proportional composition is shown in Table 1, with the following preparation steps. (1) A quantity of E-51, PPGDGE, and an equivalent quantity of the curing agent DGA were mixed in a centrifuge tube at room temperature according to the ratios shown in Table 1, and the mixture was stirred with a vortex shaker for 5 min; (2) The mixed system was sonicated in an ultrasonic cleaner for 5 min, followed by vortex shaking and stirring, this was repeated several times until the system became transparent and free of air bubbles; (3) The mixed epoxy resin system was poured into the mould for curing at 60 °C/2 h, followed by 80 °C/2 h, and then 120 °C/2 h. In this study, the thermally cured series of samples is named EPx, where “x” represents the molar ratio of E-51 to PPGDGE, e.g., “EP1-0” represents the polymer obtained by curing E-51 with PPGDEG in a 1:0 molar ratio with an equivalent amount of DGA.

### 2.3. Characterization

#### 2.3.1. Differential Scanning Calorimetry (DSC) Measurements

For glass transition temperature (T_g_) testing of the material, 3–5 mg of cured film samples were weighed in an aluminum crucible. The samples were first equilibrated at −40 °C and held for 2 min, then heated to 200 °C at 10 °C/min and held at 200 °C for 1 min in order to eliminate the thermal history. Subsequently, the samples were cooled to −40 °C at 20 °C/min and then heated again to 200 °C at 10 °C/min. The thermal properties of the fully dried material were obtained under a high purity nitrogen atmosphere at 40 mL/min.

For the non-isothermal DSC curing kinetics of the thermoplastic SMEP, 3–5 mg of ready-mixed uncured samples were weighed in a liquid crucible, and DSC tests were carried out at different ramp rates of 5, 10, 15, and 20 °C/min in the temperature range 10–300 °C, to determine the curing process of the series of thermoplastic SMEPs. Then three characteristic temperatures were determined, e.g., initial temperature (T_i_), peak temperature (T_p_), and finished temperature (T_f_) for each of the exothermic curves. Then these characteristic temperature’s data were plotted against the ramp rate, and a straight line was fitted. Using extrapolation, we could obtain the T_i_, T_p_, and T_f_ for a ramp rate of 0 °C/min, which are the pre-cure, cure, and post-cure temperatures, respectively, of the resin system. Finally, the extrapolated cure temperatures were used as the basis for determining the final cure process.

#### 2.3.2. Fourier Transform Infrared Spectroscopy (FTIR)

All the samples were tested using FTIR spectroscopy on a Nicolet 6700 (Thermo Fisher Scientific, Shanghai, China), with the potassium bromide smear method for liquid samples and the reflection method for thin film samples. Thirty-two scans were performed at a scan rate of 4 cm^−1^ during the test, and the infrared spectra were recorded in the wave number range of 4000 to 500 cm^−1^.

#### 2.3.3. Thermogravimetric Analysis (TGA) 

The difference in thermal stability of the fully dried materials was tested using a Q50 (TA Instruments, New Castle, DE, USA) at 40 mL/min in a high purity nitrogen atmosphere, during which approximately 10 mg of sample was weighed in a crucible and the temperature ramped up from 40 °C to 800 °C at a rate of 10 °C/min. The TG and DTG curves were recorded.

#### 2.3.4. Tests on Mechanical Properties

The tensile specimen is a 5B dumbbell specimen in GB/T 1040.2-2006 with three samples and then tested by a SUNS UTM6103 universal mechanical testing machine (Shenzhen Sansi Test Instrument Co., Ltd., Shenzhen, China), with a test speed of 50 mm/min.

#### 2.3.5. Dynamic Thermomechanical Analysis (DMA) and Shape Memory Effect (SME) Characterizations

The multi-frequency strain mode of a dynamic thermomechanical analyzer (DMA Q800, TA instruments, New Castle, DE, USA) can be used to obtain the dynamic variation of the mechanical properties of a material, and was used in this work to explore the dynamic processes of the energy storage moduli and loss tangents, Tan Delta, with temperature for a series of samples. The polymer film was cut into rectangular splines, each with a length, width, and thickness of 25 mm, 10 mm, and 1 mm, respectively. DMA single cantilever grippers were used during the test, using a multi-frequency strain mode with a frequency set to 1 Hz, and a temperature rise rate of 3 °C/min from −40 °C to 150 °C.

The implementation of the shape memory effect (SME) is a dynamic process accompanied by changes in temperature, so the dynamic mechanical properties of shape memory polymers with temperature need to be systematically investigated. The DMA instrument with a stretching die acts as a small stretching machine in the controlled force mode, and can accurately record the changes in strain, temperature, and stress during the implementation of the SME on the material. The polymer samples’ differences were tested using tensile grips and single cantilever grips, both in controlled force mode. For the polymer samples tested using a tensile fixture, they were cut into rectangular slices, each with a length, width and thickness of 10 mm, 5 mm, and 0.5 mm, respectively. For polymer samples tested using a single cantilever gripper, they were cut into rectangular slices, each with a length, width and thickness of 25 mm, 10 mm, and 1 mm, respectively. The specific procedure was set up as shown in Table 2.

The SME is concerned with two parameters, the rate of shape fixation *(R_f_*) and the rate of shape recovery (*R_r_*). *R_f_* is the ratio of the strain fixed after cooling and withdrawal to the strain before withdrawal, and R_r_ is the degree of return from the fixed strain to the initial strain after the final warming. The rate of R_f_ and the rate of R_r_ in the curves obtained from a single shape memory test are defined by Equations (1) and (2), respectively:(1)$$\begin{array}{c} R_{f}=\frac{\varepsilon_{f}}{\varepsilon_{0}}\times 100\%\;\;\; \end{array}$$
(2)$$\begin{array}{c} R_{r}=\frac{\varepsilon_{f}-\varepsilon_{r}}{\varepsilon_{f}}\times 100\%\;\; \end{array}$$
where *ε*_0_, *ε_f_*, and *ε_r_* represent the strain value generated after the force is applied, the strain value fixed after the force is withdrawn, and the strain value returned to after the final warming, respectively. The rate of *R_f_*(*N*) and the rate of *R_r_*(*N*) in the curves obtained from the shape memory cycle test are defined by Equations (3) and (4), respectively:(3)$$\begin{array}{c} R_{f}\left(N\right)=\frac{\varepsilon_{f}\left(N\right)}{\varepsilon_{0}\left(N\right)}\times 100\%\; \end{array}$$
(4)$$\begin{array}{c} R_{r}\left(N\right)=\frac{\varepsilon_{f}\left(N\right)-\varepsilon_{r}\left(N\right)}{\varepsilon_{f}\left(N\right)-\varepsilon_{r}\left(N-1\right)}\times 100\%\;\; \end{array}$$
where *ε*_0_(*N*), *ε_f_*(*N*), and *ε_r_*(*N*) represent the strain value generated after the force is applied at the Nth cycle, the strain value fixed after the force is withdrawn, and the strain value returned to after the final warming, respectively.

#### 2.3.6. Structure and Morphology Characterization

An atomic force microscope (AFM, Bruker Dimension Icon, Germany) was used to observe the micro-phase separation structure of the polymer samples in tap mode. For linear polymers, a quantity of sample was dissolved in DMF to form a polymer solution with a mass fraction of 0.5 wt%, then the solution was added dropwise to a clean silicon wafer and spin-coated using a homogeniser at 500 rpm for 10 s at low speed, then at 2000 rpm for 60 s at high speed, and finally the spin-coated wafer was dried in an oven at 60 °C for 24 h before AFM testing. The polymer film was cut into squares of 5 mm × 5 mm × 0.5 mm (length, width, and thickness, respectively) and adhered flat to the wafer for AFM testing. The microscopic morphology of the polymer sample fractures were imaged by scanning electron microscopy (SEM SU-70, Hitachi, Tokyo, Japan). Firstly, the polymer samples were embrittled in liquid nitrogen and then attached flush to the sample stage with conductive adhesive with the fracture side up, sprayed with gold for 80 s, and then placed in an electron microscope for observation. In addition, to observe the semi-interpenetrating network structure, the polymer samples were embrittled in liquid nitrogen, soaked in DMF at 120 °C for 12 h to etch away the linear polymer fraction, and then the remaining polymer samples were dried. The rest of the operation is the same as above.

## 3. Results and Discussion

### 3.1. Bulk Polymerization of Thermoplastic SMEPs Systems

The curing process, including curing method, curing temperature and curing time, is an important factor affecting the performance of the material. The thermal curing of epoxy resins is generally carried out in a step-wise manner to ensure an adequate reaction and to reduce material defects, mainly by using the non-isothermal differential scanning calorimeter (DSC 25, TA Instruments, New Castle, DE, USA) to determine the step-wise curing process. In this study, non-isothermal DSC testing of all of the reaction systems was carried out using four ramp rates (β), of 5, 10, 15, and 20 °C/min, to determine the two key curing processes of cure temperature and cure time, to obtain the exothermic curves at different ramp rates, as shown in Figure 2. It can be observed that the peak exothermic curve temperature decreases with increasing ramp rate, due to the fact that the reaction system does not have sufficient cure time at faster ramp rates and therefore the exothermic curve shifts to a higher temperature to compensate for the reduced time. In a DSC exothermic curve, the T_i_ is the temperature corresponding to the intersection of the tangent line of maximum slope with the baseline, the T_p_ is the temperature corresponding to the peak of the exothermic curve, and the T_f_ is the temperature corresponding to the intersection of the curve of minimum slope with the baseline. Each exothermic curve was processed using the TRIOS software to obtain its corresponding T_i_, T_p_, and T_f_ values.

From the non-isothermal DSC curves, we can see that the three characteristic temperatures (T_i_, T_p_, and T_f_) are parameters related to the temperature increase rate. In order to avoid the influence of the heating rate on the characteristic, temperatures are obtained by extrapolation for β = 0 °C/min. The three characteristic temperatures are the reaction temperatures at a constant temperature, and the curing of epoxy polymers generally takes place at a constant temperature. The extrapolated characteristic temperatures match the actual curing process of the epoxy polymer. As shown in Figure 3, the characteristic temperatures for each system were plotted against the heating rates and straight lines were fitted. The linearity of each fitted line was good (R^2^ > 0.98), so the extrapolated characteristic temperatures at a constant temperature (β = 0 °C/min) were reliable. The extrapolated characteristic temperatures for each system are summarized in Table 3, where it can be found that all systems have an extrapolated start temperature around 44 °C, an extrapolated peak temperature around 80 °C, and an extrapolated final temperature between 103 and 126 °C. Therefore, the curing process for all systems was set at 60 °C/2 h, followed by 80 °C/2 h, and then 120 °C/2 h, after comprehensive consideration.

### 3.2. Structural Analysis of Thermoplastic SMEP

The series of EPx samples were characterized using FTIR to determine the molecular structure of the polymer samples by the characteristic IR absorption peaks of the different functional groups. Figure 4 shows the FTIR spectra of EP12-1 before and after curing with monomers DGA, E-51, and PPGDGE, where the more obvious absorption peaks at 1510 cm^−1^ and 1110 cm^−1^ correspond to C=C bonds in the benzene ring group of E-51, and C-O-C in the chain segment of PPGDEG, respectively. Strong absorption peaks at 910 cm^−1^ can also be found for E-51, PPGDGE, and uncured EP12-1, which are characteristic IR absorption peaks for epoxy groups. However, the absorption peak of the EP12-1 sample disappeared almost completely, indicating that the epoxy group reacted almost completely during the heat curing process, and that the target product was prepared successfully. In addition, the DGA sample had an obscure double peak near 3300 cm^−1^, which corresponds to two N-H bonds in the primary amine group in its molecular structure, while these two absorption peaks disappeared in the cured EP12-1 sample, indicating that the primary amine group in the DGA had completely reacted with the epoxy group during the heat curing process, further demonstrating the successful preparation of the target product.

The molecular structures of all samples were further investigated, and the FTIR spectra of the series of EPx samples are shown in Figure 5. It can be observed that the characteristic absorption peak of the epoxy group at 910 cm^−1^ disappears almost completely for all samples. This result fully demonstrates that the EPx system can be cured almost completely in response to the curing process. In addition, all samples show a broad absorption peak around 3370 cm^−1^, which is a stretching vibration of the hydroxyl group. On the one hand, this group is ascribed to the hydroxyl at the chain end of the DGA molecule in the product monomer, on the other hand, it could result from the reaction of the epoxy group with the N-H group during the thermal curing of the epoxy and amine. In addition, the appearance of the C-H stretching vibration peak near 2900 cm^−1^, the C=C vibration peak in the benzene ring group at 1510 cm^−1^, and the C-O-C stretching vibration peak at 1110 cm^−1^ in all samples, also indicates that the rigid benzene ring group in E-51 and the flexible ether bond in PPGDGE were successfully introduced into the polymer chain segment, and the target products were successfully synthesized.

### 3.3. Thermal Performance Analysis of Thermoplastic SMEP

The T_g_ is important for shape memory polymers as it is an important transition unit that controls the temporary shape change in shape memory polymers. In this study, T_g_ is the switching temperature for the shape memory effect in the series of EPx samples. In the implementation of shape memory, the polymer is manually deformed by external forces above T_g_ in the highly elastic state, and then cooled to below T_g_ in the glassy state when the external forces are removed and the temporary shape is ‘frozen’. Therefore, this work systematically investigates the T_g_ of a series of samples by using DSC, as shown in Figure 6. It can be observed that the T_g_ of the series of samples decreases as the PPGDGE content increases. This is mainly due to the fact that PPGDEG is a flexible C-O-C structure, whereas E-51 is a rigid structure with a large number of benzene ring groups. The introduction of this C-O-C structure into the polymer main chain improves the flexibility of the polymer chain segments, which leads to a decrease in T_g_.

The thermal decomposition of polymeric materials at high temperatures leads to the destruction of their chemical structure and composition, which affects the properties of the materials. The TGA test provides a clear indication of the thermal decomposition temperature of the polymer material, monitors the thermal decomposition process of the material, and provides an important basis for obtaining thermal stability parameters for the range of temperatures used in the processing, production, and application of the material. As can be seen from Figure 7, the TGA curves have only one weight loss plateau, indicating that there is mainly only one stage of weight loss in the polymer samples. The temperature at which there is >5 wt% weight loss is generally defined as the initial decomposition temperature, and the initial thermal decomposition temperature for the series of samples ranged from 337–355 °C. This excellent thermal stability of the material is due to the large number of conjugated benzene ring structures in the polymer molecular chain segments. At the same time, the initial thermal decomposition temperatures of the series of samples are well above their T_g_, indicating that all systems have excellent thermal stability within the T_g_ transition range, and high temperatures should not affect the normal use of the material at all. As can be seen from the local magnification, the initial thermal decomposition temperature of the series of samples decreases slightly as the PPGDGE content of the system increases, which is mainly due to the increase in the content of flexible chain segments and the decrease in the content of rigid chain segments in the system. Meanwhile, the DTG curves show that the maximum thermal weight loss rate for the series of samples is around 390 °C, which is not significantly affected by the addition of PPGDGE.

### 3.4. Mechanical Properties Analysis of Thermoplastic SMEP

#### 3.4.1. Tensile Mechanical Properties

The mechanical properties of the EPx series were assessed using tensile tests at room temperature to obtain the fracture stress and strain. As can be seen from Figure 8, the fracture stresses for the series of samples ranged from 7 MPa to 34 MPa, and the fracture strains ranged from 30% to 700%. The fracture stress of a polymer is positively proportional to the content of rigid groups in its main chain, while the fracture strain of a polymer is positively proportional to the content of flexible groups in its main chain. The EP1-0 system has the highest fracture stress due to the rigid benzene ring group, mainly E-51, in the main chain of the EP1-0 molecule, which also results in a fracture strain of 31%. When PPGDGE was introduced into the system, the fracture strain of the material increased significantly, and continued to increase as the PPGDGE content increased. The introduction of the flexible ether bonding structure in PPGDGE greatly improved the flexibility of the polymer chain segments and the deformability of the material increased significantly. Of course, the introduction of more and more flexible chain segments into the polymer main chain inevitably leads to a reduction in the fracture strength of the material.

#### 3.4.2. Dynamic Mechanical Properties

Figure 9a demonstrates the high modulus of the series of EPx samples in the low temperature range, with a dramatic drop in the modulus as the temperature rises until a stable plateau region occurs. This is the transition of the material from a high modulus glassy state at low temperatures to a low modulus rubbery state at high temperatures, where the ability of the polymer’s molecular chain segments to move increases as the temperature rises. In addition, the energy storage modulus of the series of samples at room temperature (25 °C) varied between 480 MPa for EP8-1 and 2.2 GPa for EP1-0, showing a pattern of decreasing with increasing PPGDGE content, which is mainly due to the fact that the higher E-51 content, with a rigid benzene ring group structure, the more rigid the material becomes. The loss factor Tan Delta is the ratio of the energy storage modulus to the loss modulus of the material, and its peak temperature is the T_g_ of the material. It can be seen from Figure 9b that as the content of PPGDGE increases, the peak Tan Delta temperature of the series of samples moves towards lower temperatures, which is consistent with the trend of T_g_ obtained from the DSC test. In addition, it was found that the T_g_ of the series of samples obtained from the DMA test was about 10 °C higher than that obtained from the DSC test. The DMA test is relatively more accurate in determining T_g_ based on modulus changes. Therefore, during subsequent shape memory tests, the T_g_ is determined by the DMA test.

### 3.5. Shape Memory Performance of Thermoplastic SMEP

The shape memory curves in Figure 10a–e are made up of three processes: warming up and stretching, cooling down and withdrawing to fix the deformation, warming up and recovery to deformation, divided into two main stages: the shape fixation stage and the shape recovery stage. The DMA tensile shape memory test controls the material strain between 100% and 150%. It is rare to assess the shape memory performance of SMEP at such high strains, as SMEP reported in the literature have difficulty remembering temporary shapes with strains of up to 100%. From Figure 10, it can be seen that the shape fixation rates of the series of EPx samples are all close to 100%, and the shape recovery rates gradually increase from 87.98% to over 95% as the PPGDGE content increases. In terms of the polymer chain structure, the rigid benzene ring structure in E-51 acts as the hard (or stationary) phase in the polymer chain, while the DGA, with flexible side chains, and the PPGDGE, with flexible ether bond structures, act as the soft (or reversible) phase in the polymer chain. When the amount of reversible phase in the samples was low, the material did not have sufficient ‘driving force’ in the strain recovery process, resulting in less than 90% shape recovery for the EP1-0 and EP20-1 samples, but as the amount of reversible phase PPGDGE increased, the recovery rate gradually increased to 97.76% for EP12-1. It is important to note that the higher the reversible phase content, the better the shape memory recovery rate. The shape recovery rate of EP8-1 decreases significantly from 97.76% to 94.64% compared to EP12-1, which has a higher content of reversible phases, and there is an intermediate optimum for the proportion of reversible phases to improve the R_r_ of the material.

### 3.6. Structure and Morphology of Thermoplastic SMEP

AFM can obtain microscopic morphological information and the micro-phase separation structure of the polymer sample’s surface. The 2D phase diagram in Figure 11a shows that the series of EPx samples have a bright and dark two-phase structure, with the brighter part being the hard part of the SMEP and the darker part being the soft part. At the same time, the AFM can simulate a 3D image of the nanoscale morphology of the surface of the material, and the raised and concave parts of the 3D height map in Figure 11b are the hard and soft micro-segments of the samples, respectively. All samples showed a relatively homogeneous micro-phase separation structure. From the 3D height map, it is clear that in the samples with a high E-51 content, such as EP1-0 and EP20-1, the hard segments are particularly prominent and concentrated, and the micro-phase separation is not high, so the shape memory recovery rate is less than 90% for both samples. As the E-51 content decreases and the PPGDGE increases, the hard and soft segments become more uniformly distributed, especially in the EP12-1 sample where the hard segment micro-regions are very uniformly dispersed in a continuous soft segment matrix with a high degree of micro-phase separation, and therefore have the best shape memory performance.

The series of EPx samples were fractured in liquid nitrogen to obtain the morphology of their fracture surfaces, as shown in Figure 12. It can be seen that the EP1-0 samples are relatively smooth and neat, with unidirectional crack extension and less branching, typical of a brittle fracture morphology. With the addition of PPGDGE, the fracture surface of the samples became rougher, showing a lamellar structure and a large number of branches in the cracks, which is an indication of the improved toughness of the material, and the introduction of flexible PPGDE into the polymer structure significantly enhanced the toughness of the material, which corresponds to the tensile test.

### 3.7. Recycling Performance of Thermoplastic SMEP

The range of EPx samples can all be dissolved in DMF. For example, EP20-1 can be dissolved in DMF in an oil bath at 80 °C for 2 h. After drying the DMF, a new film can be obtained, as shown in Figure 13. The material can be reprocessed by solvent casting, making SMEP polymers not limited to the traditional casting and molding method. Since the molecular structures in this study were designed using bifunctional raw materials, the polymers prepared should theoretically be thermoplastic linear structures with no chemical cross-linking [31]. The solubility of the series of samples in DMF also demonstrates the feasibility of this structural design, and the experiments were successful in producing a predetermined structure of epoxy shape-memory polymers. The linear polymer chain structure also allows EPx to be converted from a rubbery state to a melt at high temperatures for melt reprocessing, which gives the series of EPx samples the potential for injection molding, extrusion, and especially FDM printing molding applications. These properties make these thermoplastic shape memory epoxy polymers good candidates for potential 3D printing, due to their good adhesiveness and low volume shrinkage.

## 4. Conclusions

In this study, a series of thermoplastic epoxy polymers with good shape memory effects have been prepared by bulk polymerization using commonly used epoxy resins and mono-amine diethylene glycol amines (DGA) as raw materials. All the samples show good shape fixation rates, close to 100%, and shape recovery rates, as high as 97.76%. The non-isothermal DSC, and FTIR, demonstrate the successful polymerization of the target products and show the reliable curing process. The thermal performance tests, by DSC and TGA, show that the T_g_ decreases with increasing PPGDGE content, and the samples have a constant good thermal stability. The tensile mechanical properties, tested using a universal tensile tester, demonstrate that the incorporation of PPGDGE provides excellent fracture strain. The addition of PPGDEG contributes to a significant increase in the degree of micro-phase separation and thus enhances the shape recovery of the samples. Unlike conventional thermoset epoxy polymers, this work has prepared thermoplastic SMEPs with a linear molecular chain structure, which allows the materials to be easily recycled and reprocessed by dissolution or melting. It breaks the limitation of epoxy polymer only being prepared with cast molding. Thus, the materials are potentially good candidates for 3D printing.

## Figures and Tables

**Figure 1 polymers-15-00809-f001:**
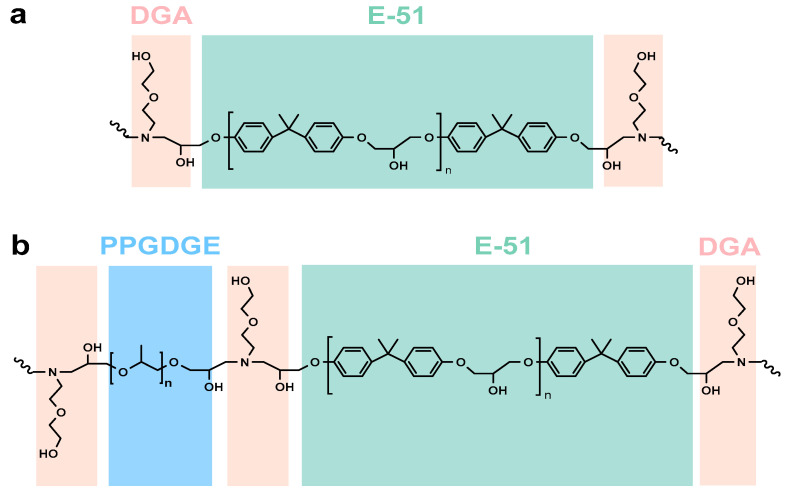
Chemical structure of thermoplastic epoxy shape memory polymer (**a**) without PPGDGE, or with PPGDGE (**b**).

**Figure 2 polymers-15-00809-f002:**
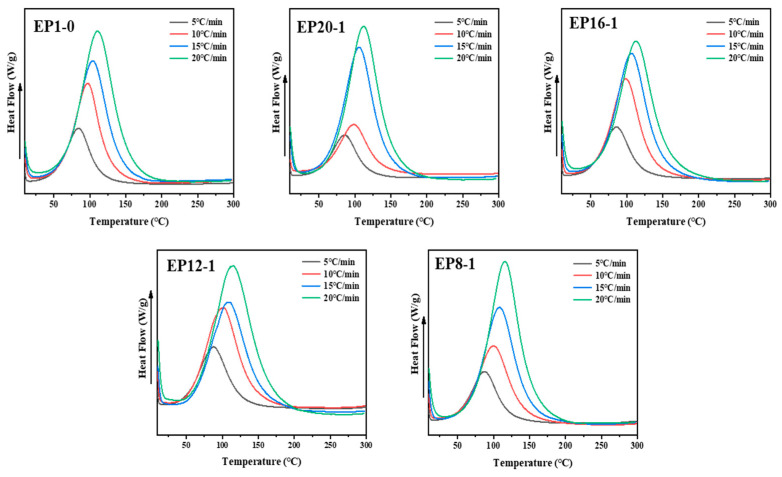
The DSC curves of EPx systems under four different heating rates.

**Figure 3 polymers-15-00809-f003:**
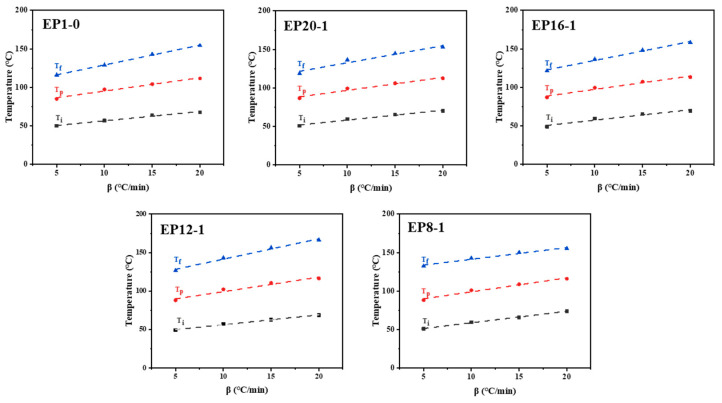
Linear fitting diagram of characteristic temperature-heating rates of the EPx systems.

**Figure 4 polymers-15-00809-f004:**
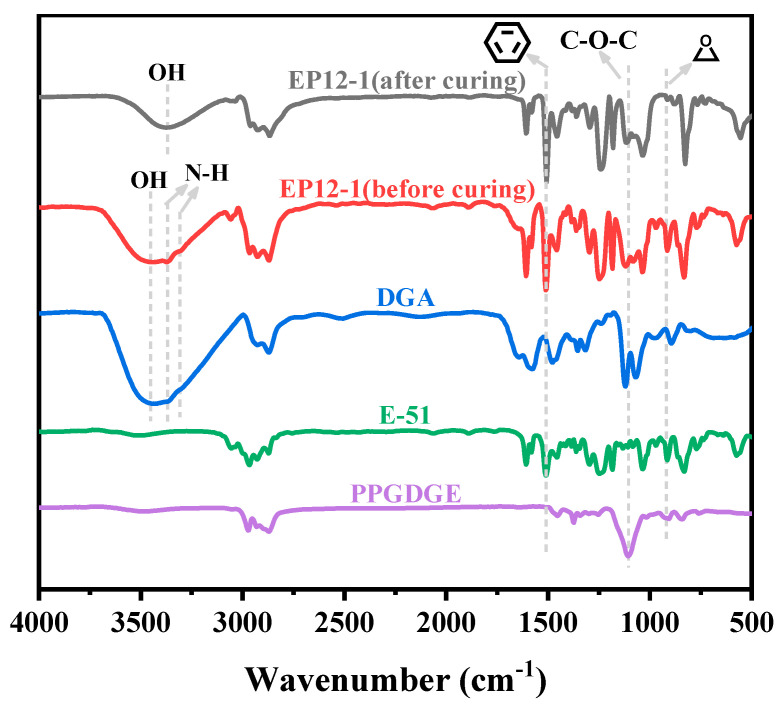
FTIR spectra of DGA, E-51, PPGDGE, and EP12-1 before and after curing.

**Figure 5 polymers-15-00809-f005:**
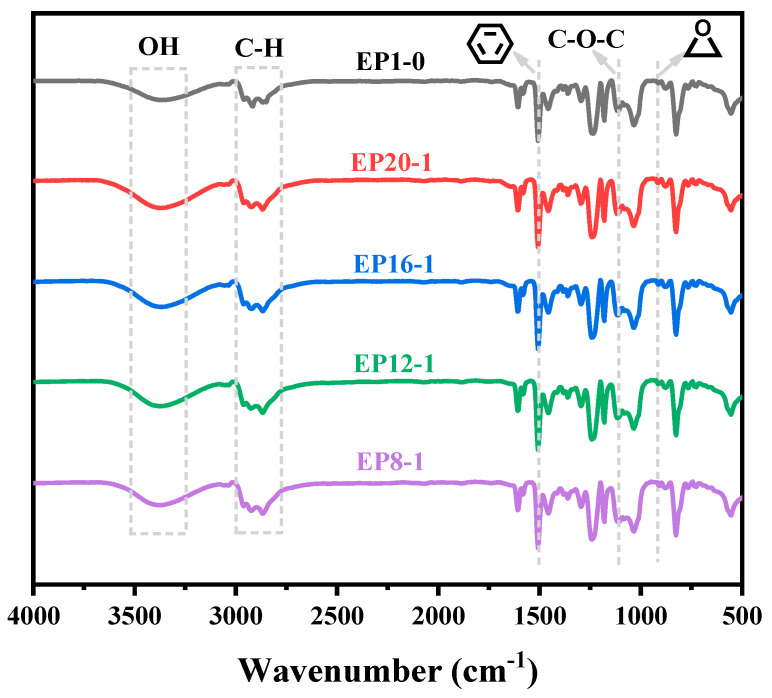
FT-IR spectra of EPx samples.

**Figure 6 polymers-15-00809-f006:**
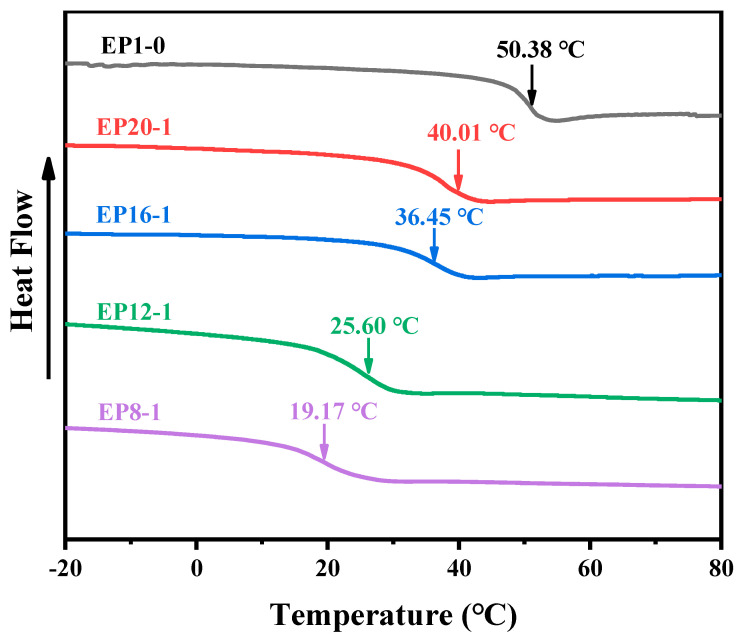
The second heating curve of EPx samples.

**Figure 7 polymers-15-00809-f007:**
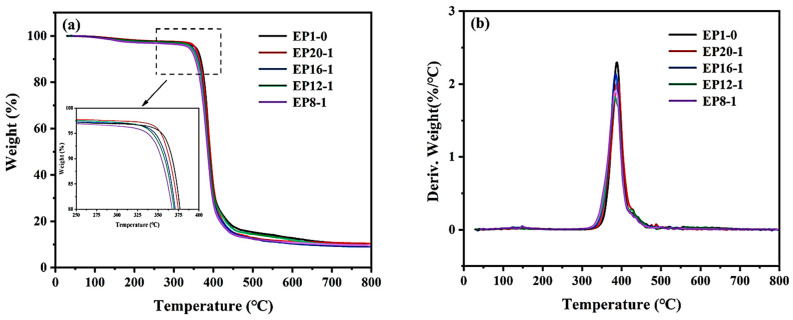
TGA curves (**a**) and DTG curves (**b**) of EPx samples.

**Figure 8 polymers-15-00809-f008:**
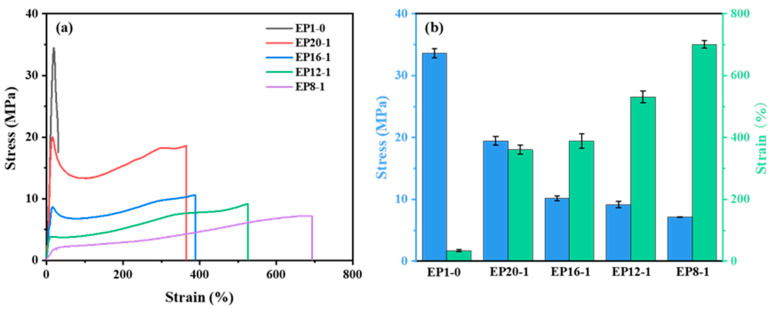
Strain-Stress curves (**a**) and statistics of Strain-Stress curves (**b**) of EPx samples.

**Figure 9 polymers-15-00809-f009:**
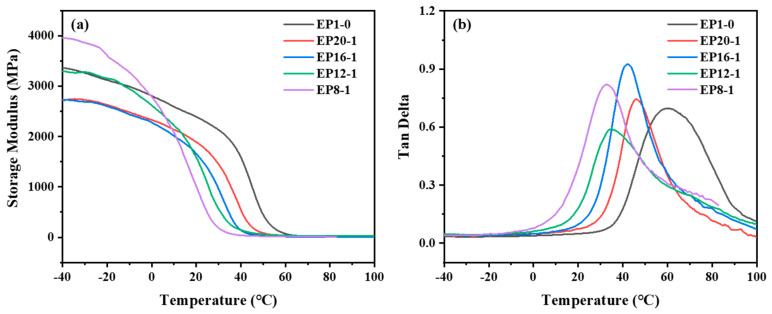
The storage modulus curves (**a**) and the loss factor curves (**b**) of EPx samples.

**Figure 10 polymers-15-00809-f010:**
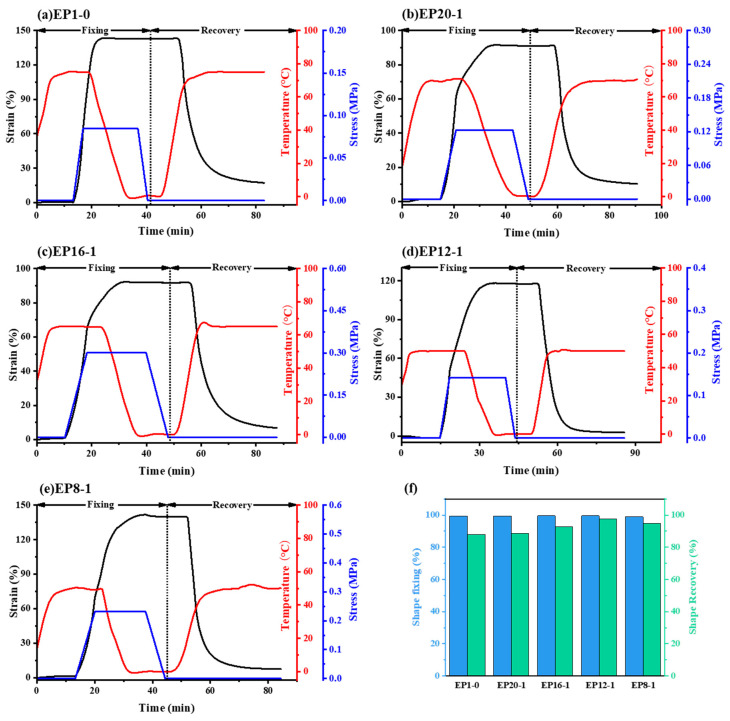
Shape memory curves (**a**–**e**) and statistics of shape memory properties (**f**) of EPx samples.

**Figure 11 polymers-15-00809-f011:**
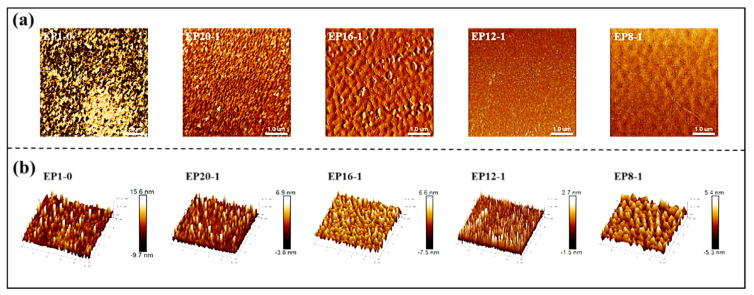
AFM 2D phase images (**a**) and 3D height images (**b**) of EPx samples.

**Figure 12 polymers-15-00809-f012:**
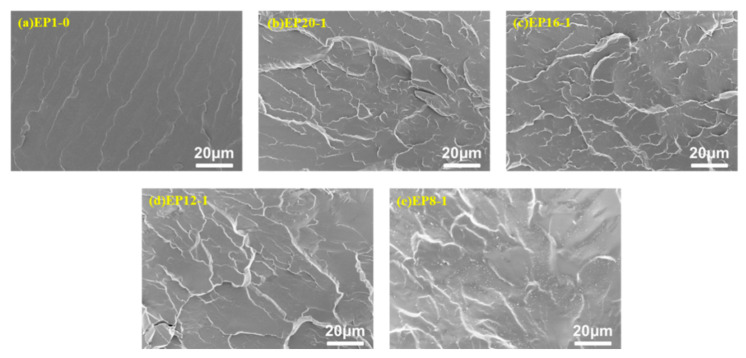
SEM images of the brittle fractured surfaces of EPx samples fractured in liquid nitrogen. (**a**) EP1-0, (**b**) EP20-1, (**c**) EP16-1, (**d**) EP12-1, (**e**) EP8-1.

**Figure 13 polymers-15-00809-f013:**
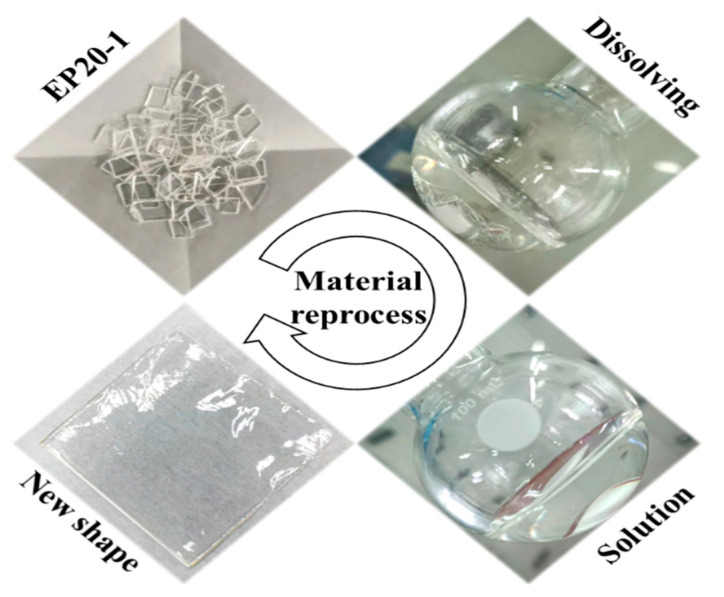
Recycling performance of EP20-1.

**Table 1 polymers-15-00809-t001:** Composition of thermoplastic epoxy shape memory polymer EPx series.

Samples	E-51 (g)	PPGDGE (g)	DGA (g)
EP1-0	3.92	-	1.05
EP20-1	3.73	0.30	1.05
EP16-1	3.69	0.37	1.05
EP12-1	3.62	0.48	1.05
EP8-1	3.48	0.69	1.05

**Table 2 polymers-15-00809-t002:** Specific procedures of DMA for SME.

Step 1	increase the temperature at a rate of 10 °C/min to 20 °C above T_g_ of the sample
Step 2	isothermal for 10 min
Step 3	apply force at the appropriate force rate to deform the sample
Step 4	isothermal for 5 min
Step 5	ramp to 0 °C at a rate of 10 °C/min
Step 6	isothermal for 10 min
Step 7	withdraw force to 0 N at the same rate as step 3
Step 8	isothermal for 5 min
Step 9	increase the temperature to 20 °C above the T_g_ at a rate of 10 °C/min
Step 10	isothermal for 30 min

**Table 3 polymers-15-00809-t003:** Extrapolated characteristic temperatures of the EPx systems.

Systems	Extrapolated Characteristic Temperatures		
	T_i_/°C	T_p_/°C	T_f_/°C
EP1-0	44.39	77.91	103.37
EP20-1	45.38	80.06	110.54
EP16-1	43.85	80.13	110.44
EP12-1	43.51	80.49	115.11
EP8-1	43.90	80.88	126.16

## Data Availability

Not applicable.

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
