# Peer review of "Bulk Polymerization of Thermoplastic Shape Memory Epoxy Polymer for Recycling Applications"

_polymers, 2023, doi:10.3390/polym15040809_

Round 1

Reviewer 1 Report

Dear Author,

The article describing the shape memory behavior of thermoplastic using polyethylene is really interesting to read and any readers will enjoy reading it. Here are my comments:

(1) Please add more work done in this field and add more cited work in introduction. Intoduction is too short and was not able to focus what has been done before and why your study is more important.

(2) For section 3.4.1 describing mechanical properties, please what does stress strain curve tell you about structure-property relationship. What is the origin of different stress strain pattern with different composition.

Author Response

Comments of Reviewer 1.

The article describing the shape memory behavior of thermoplastic using polyethylene is really interesting to read and any readers will enjoy reading it. Here are my comments:

  • Please add more work done in this field and add more cited work in introduction. Introduction is too short and was not able to focus what has been done before and why your study is more important.

Reply: Thanks for your suggestions.

   We had revised the introduction. More work in this field and more literature are cited in the revisions. Such as

(2) For section 3.4.1 describing mechanical properties, please what does stress strain curve tell you about structure-property relationship. What is the origin of different stress strain pattern with different composition.

Reply: Thanks for your suggestions.

     We had revised the discussion on mechanical properties. The structure-property relationship were discussed carefully. E.g. “ These results confirmed that the fracture stress of a polymer was positively proportional to the content of rigid groups in its main chain, while the fracture strain of a polymer was positively proportional to the content of flexible groups in its main chain. 

   Additionally, we had also explained the reasons for different composition. E.g. “The EP1-0 system has the highest fracture stress due to the rigid benzene ring group, mainly E-51, in the main chain of the EP1-0 molecule, which also results in a fracture strain of 31%. When PPGDGE was introduced into the system, the fracture strain of the material increased significantly and continued to increase as the PPGDGE content increased in EP20-1, EP16-1, EP-12-1 and EP8-1. ”

Reviewer 2 Report

In this paper, thermoplastic Shape memory epoxy has been polymerized and mechanical, thermal and shape memory properties have been comprehensively investigated. The results and topic of the article are interesting, but some parts of the manuscript must be completed to be published in the journal.

The novelty of the article should be presented clearly. Is this combination used for the first time?

Also, what is the use of thermoplastic shape memory epoxy and what are the advantages and disadvantages of these materials compared to thermoplastics?

The introduction is written very superficially and briefly. Also, the sources reviewed in the introduction are very few. The introduction should be rewritten entirely. The introduction of shape memory polymers, thermoplastic shape memory epoxy (their advantages and applications compared to other shape memory materials) and their preparation methods should be added in the introduction. It is suggested to use these references (“4D Printing-Encapsulated Polycaprolactone–Thermoplastic Polyurethane with High Shape Memory Performances” --- “A New Strategy for Achieving Shape Memory Effects in 4D Printed Two-Layer Composite Structures” --- “4D printing of PET-G via FDM including tailormade excess third shape”).

What is the basis of the selected combinations (Table 1)? Is it just right and wrong or does it have sources and theoretical support?

The DSC and DMTA section is long and should be summarized. It is suggested to provide programming parameters in a table.

Has the shape memory test been done in one cycle? Is it suggested to check its cyclical behavior?

Reproducibility of mechanical properties and shape memory effect with how many samples have been done?

Most of the results sections are just reports of experimental test, and in-depth analysis should be added to them.

Hasn't the recovery stress and stress relaxation been checked for this material?

Author Response

In this paper, thermoplastic Shape memory epoxy has been polymerized and mechanical, thermal and shape memory properties have been comprehensively investigated. The results and topic of the article are interesting, but some parts of the manuscript must be completed to be published in the journal.

  1. The novelty of the article should be presented clearly. Is this combination used for the first time?

Reply: thanks for your comments and suggestions.

We had revised the manuscript carefully. The novelty which was presented in introduction section is that : “a novel thermoplastic shape memory epoxy resin with high thermal stability and low shrinkage are prepared with bisphenol A epoxy resin E-51, flexible structured PPGDGE and monoamine curing agent DGA.”

  1. Also, what is the use of thermoplastic shape memory epoxy and what are the advantages and disadvantages of these materials compared to thermoplastics?

Reply: thanks.

      In the introduction, we had pointed out the disadvantages of common SMEPs and the advantages of thermoplastic shape memory epoxy : e.g. “However, the majority of SMEPs in the current study are cross-linked in a three-dimensional network structure, leading to a general problem of low fracture strain in their mechanical properties, and limited to the traditional casting and forming process in terms of manufacturing and processing, which limits the further development and application of the material. Unlike conventional thermoset SMEP, thermoplastic SMEP have good recyclability, repairing and shape memory properties[25] and can be used in a variety of applications,such as smart coatings[26].

  1. The introduction is written very superficially and briefly. Also, the sources reviewed in the introduction are very few. The introduction should be rewritten entirely. The introduction of shape memory polymers, thermoplastic shape memory epoxy (their advantages and applications compared to other shape memory materials) and their preparation methods should be added in the introduction.

Reply: thanks for your good comments.

   The introduction was rewritten carefully. More sources were reviewed and cited. We had presented the advantages and applications of thermoplastic shape memory epoxy and the common preparation methods were added in the introduction.

  1. It is suggested to use these references (“4D Printing-Encapsulated Polycaprolactone–Thermoplastic Polyurethane with High Shape Memory Performances” --- “A New Strategy for Achieving Shape Memory Effects in 4D Printed Two-Layer Composite Structures” --- “4D printing of PET-G via FDM including tailormade excess third shape”).

Reply: thanks for your good suggestions.

    We had added the possible applications of thermoplastic shape memory epoxy in the introduction by use the recommended references, e.g. “ they can be used in a variety of applications, such as 4D printing[21-23], smart coatings[26].”

  1. What is the basis of the selected combinations (Table 1)? Is it just right and wrong or does it have sources and theoretical support?

Reply: thanks.

 The selected combinations in Table 1 are designed according to reaction between epoxy group with -NH. To obtained thermoplastic shape memory epoxy, the ratio is kept to 1:1 in this work. 

  1. The DSC and DMTA section is long and should be summarized. It is suggested to provide programming parameters in a table.

Reply: thanks for your good suggestions.

We had revised the procedure of DSC test for For glass transition temperature and the non-isothermal DSC curing kinetics of the thermoplastic SMEP,

The specific procedure of DMTA is summarized in Table 2. Most of the programming parameters are presented in table 2.

  1. Has the shape memory test been done in one cycle? Is it suggested to check its cyclical behavior?Reproducibility of mechanical properties and shape memory effect with how many samples have been done?

Reply: thanks

     In this study, the shape memory test have been done in one cycle since The DMA instrument can accurately record the changes in strain, temperature and stress during the implementation of the SME on the material. This test method was widely used in many literatures.

  1. Most of the results sections are just reports of experimental test, and in-depth analysis should be added to them.

 Reply: thanks. we had revised the manuscript according to the suggestions of reviewers. Some in-depth analysis was added in the introduction and discussions.

9.Hasn't the recovery stress and stress relaxation been checked for this material?

Round 2

Reviewer 2 Report

Accept